# Tuning magnetoelectricity in a mixed-anisotropy antiferromagnet

Ellen Fogh [1,2] ✉, Bastian Klemke [3], Manfred Reehuis[3], Philippe Bourges [4], Christof Niedermayer [5], Sonja Holm-Dahlin [5,6], Oksana Zaharko [5], Jürg Schefer[5], Andreas B. Kristensen[2], Michael K. Sørensen[2], Sebastian Paeckel [3,7], Kasper S. Pedersen [8], Rasmus E. Hansen[9], Alexandre Pages[1], Kimmie K. Moerner[2], Giulia Meucci[2], Jian-Rui Soh [1], Alessandro Bombardi [10], David Vaknin [11], Henrik. M. Rønnow [1], Olav F. Syljuåsen [12], Niels B. Christensen [2] ✉ & Rasmus Toft-Petersen [2,13] ✉

Control of magnetization and electric polarization is attractive in relation to tailoring materials for data storage and devices such as sensors or antennae. In magnetoelectric materials, these degrees of freedom are closely coupled, allowing polarization to be controlled by a magnetic field, and magnetization by an electric field, but the magnitude of the effect remains a challenge in the case of single-phase magnetoelectrics for applications. We demonstrate that the magnetoelectric properties of the mixed-anisotropy antiferromagnet $LiNi_{1-x}Fe_xPO_4$ are profoundly affected by partial substitution of $Ni^{2+}$ ions with $Fe^{2+}$ on the transition metal site. This introduces random site-dependent single-ion anisotropy energies and causes a lowering of the magnetic symmetry of the system. In turn, magnetoelectric couplings that are symmetry-forbidden in the parent compounds, $LiNiPO_4$ and $LiFePO_4$, are unlocked and the dominant coupling is enhanced by almost two orders of magnitude. Our results demonstrate the potential of mixed-anisotropy magnets for tuning magnetoelectric properties.

Utilizing the magnetoelectric (ME) effect to electrically control magnetic states has far-reaching prospects in next-generation electronics[1,2]. Currently realised applications are based on heterostructure composites[3], combining layers with distinct bulk properties. Proposals for application of the ME effect in heterostructures are abound, including electric-field control of skyrmions[4], magnetoelectric

spin-orbit logic devices[5,6], medical implants[7–9] and low-power-consumption ME random access memory[10–12]. As many of these proposals involve distinct ME layers, a fundamental understanding of the properties of single-phase magnetoelectrics is pivotal to their realization. While our understanding of the underlying mechanisms has been greatly improved since the discovery of the ME effect, the relatively

[1]Laboratory for Quantum Magnetism, Institute of Physics, École Polytechnique Fédérale de Lausanne (EPFL), CH-1015 Lausanne, Switzerland. [2]Department of Physics, Technical University of Denmark, DK-2800 Kongens Lyngby, Denmark. [3]Helmholtz-Zentrum Berlin für Materialien und Energie, D-14109 Berlin, Germany. [4]Université Paris-Saclay, CNRS, CEA, Laboratoire Léon Brillouin, 91191 Gif-sur-Yvette, France. [5]Laboratory for Neutron Scattering and Imaging, Paul Scherrer Institute, Villigen CH-5232, Switzerland. [6]Nano-Science Center, Niels Bohr Institute, University of Copenhagen, DK-2100 Copenhagen Ø, Denmark. [7]Department of Physics, Arnold Sommerfeld Center for Theoretical Physics (ASC), Munich Center for Quantum Science and Technology (MCQST), Ludwig-Maximilians-Universität München, 80333 München, Germany. [8]Department of Chemistry, Technical University of Denmark, DK-2800 Kongens Lyngby, Denmark. [9]Department of Photonics Engineering, DK-2800 Kongens Lyngby, Denmark. [10]Diamond Light Source Ltd., Harwell Science and Innovation Campus, Didcot, Oxfordshire OX11 0DE, UK. [11]Ames National Laboratory and Department of Physics and Astronomy, Iowa State University, Ames Iowa 50011, USA. [12]Department of Physics, University of Oslo, P. O. Box 1048, Blindern N-0316 Oslo, Norway. [13]European Spallation Source ERIC, P.O. Box 176, SE-221 00 Lund, Sweden. ✉e-mail: ellen.fogh@epfl.ch; nbch@fysik.dtu.dk; rasp@fysik.dtu.dk

weak ME couplings are lingering barriers for applicability of single-phase magnetoelectrics.

The ME properties of a given single-phase material are a consequence of the magnetic point group symmetry inherent to its magnetically ordered state[13,14]. More specifically, the absolute and relative orientation of the ordered moments dictate the non-zero elements of the ME tensor describing the coupling between electric and magnetic degrees of freedom[14,15]. Mixing magnetic ions with incompatible, or mismatched, single-ion anisotropies gives rise to what can be thought of as a composite on the atomic level. This random site-dependent anisotropy in combination with the inter-species

exchange interaction creates frustration in the system and may result in what is known as an oblique antiferromagnetic phase. Here, the ordered moments are oriented away from any of the easy axes observed in the stoichiometric compounds[16–18].

A well-known family of isostructural magnetoelectric has chemical formula $LiMPO_4$ ($M$ = Mn, Fe, Co, Ni) and space group $Pnma$ (No. 62)[19] with the crystallographic unit cell illustrated in Fig. 1a. The compounds, $LiNiPO_4$ ($S=1$)[20–22] and $LiFePO_4$ ($S=2$)[23,24] order antiferromagnetically at 20.8 K and 50 K, respectively. Below their Néel temperatures, they display similar commensurate spin structures except for the orientation of the magnetic moments, which are predominantly along the

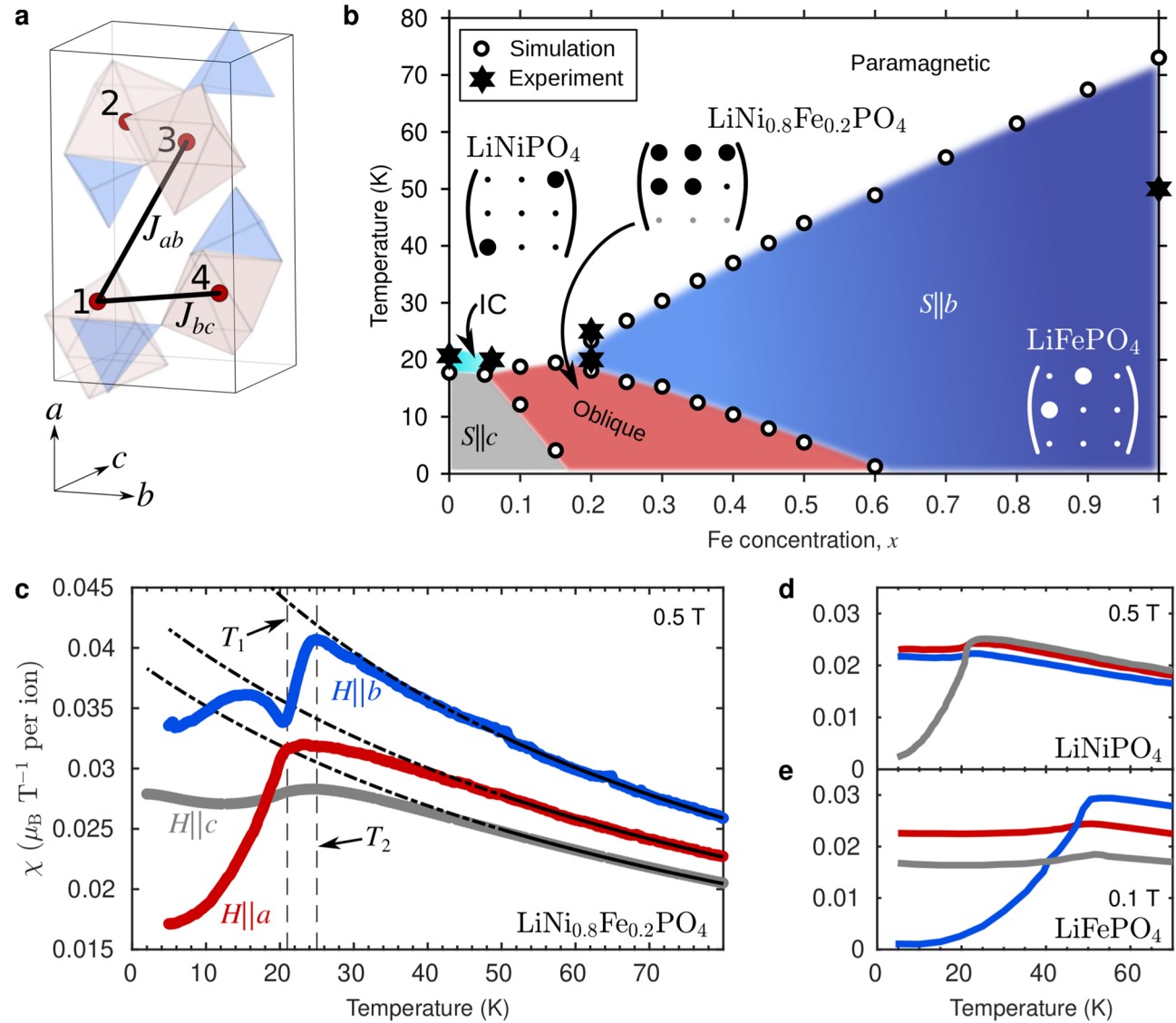

**Fig. 1 | Unit cell, (x, T) phase diagram and magnetic susceptibility of $LiNi_{1-x}Fe_xPO_4$.** **a** Crystallographic unit cell of $LiMPO_4$ with four magnetic ions (red numbered spheres) and the two most important exchange paths, $J_{bc}$ and $J_{ab}$, shown. The $MO_6$ octahedra and $PO_4$ tetrahedra are illustrated with red and blue shading, respectively. **b** (x, T) phase diagram constructed from experimental data and simulation. The open circles correspond to phase transitions observed in the simulated specific heat. Filled stars represent phase transitions detected in magnetic susceptibility and neutron diffraction experiments for samples with x = 0, 0.06, 0.20 and 1. Both simulations and experiments reveal three phases: Commensurate phases with $S||c$ (grey shading) and $S||b$ (blue shading) are seen at small and large x, respectively, while an oblique phase is present in the range $0.1 < x < 0.6$ (red shading). For each phase, the observed form of the magnetoelectric tensor at low temperature is indicated. The gradient of the blue shading illustrates that the

ordered moment along $b$, $\langle S||b \rangle$, decreases when decreasing $x$ while $\langle S||a \rangle \approx \langle S||c \rangle = 0$. For small $x$ there exists an incommensurate (IC) phase in a narrow temperature interval above the commensurately ordered phase (cyan shading)[21,22]. **c** Magnetic susceptibility of $LiNi_{0.8}Fe_{0.2}PO_4$ measured with 0.5 T applied along the three crystallographic axes; $a$ (red curve), $b$ (blue curve) and $c$ (grey curve). Fits to the Curie-Weiss law are shown with black lines. The solid parts of the lines indicate the fitted interval (50–300 K) and the dash-dotted parts are extrapolations to lower temperatures. The vertical dashed lines mark transitions at 21 and 25 K. **d, e** Corresponding susceptibilities for $LiNiPO_4$ and $LiFePO_4$, measured with 0.5 and 0.1 T respectively. Reprinted with permission from refs. 32 and 54. Copyright (2023) by the American Physical Society. Error bars in all panels are smaller than symbol sizes.

crystallographic $b$ and $c$ axes for respectively LiFePO$_4$ and LiNiPO$_4$. In LiNiPO$_4$ there exists in addition an incommensurate phase in a narrow temperature interval just above the Néel temperature[21,22]. The static and dynamic properties of Li$M$PO$_4$ are well-described by the spin Hamiltonian

$$\hat{\mathcal{H}} = \sum_{\langle i,j \rangle} J_{ij} \mathbf{S}_i \cdot \mathbf{S}_j + \sum_{i,\alpha} D_i^\alpha \left( S_i^\alpha \right)^2, \qquad (1)$$

where the first sum accounts for the exchange interactions of magnitude $J_{ij}$ between spins on sites $i$ and $j$. The second sum over all sites $i$ and three crystallographic directions, $\alpha = \{a,b,c\}$, reflects single-ion anisotropy energies, parameterized by the vector $\mathbf{D} = (D^a, D^b, D^c)$. This term is responsible for the distinct ordered moment direction selected upon ordering in stoichiometric LiNiPO$_4$[25] and LiFePO$_4$[26].

Here, we explore chemical tuning of mixed-anisotropy antiferromagnets as a novel route for tailoring the properties of single-phase magnetoelectrics. We have employed magnetic susceptibility and pyrocurrent measurements, neutron diffraction and Monte Carlo simulations to investigate the $(x, T)$ phase diagram of LiNi$_{1-x}$Fe$_x$PO$_4$ (Fig. 1b). We observe three commensurate magnetic phases with propagation vector $\mathbf{k} = 0$. At low temperature and for $x < 0.2$, the spins order along $c$ like in LiNiPO$_4$. For $x > 0.6$, the spins order along $b$ like in LiFePO$_4$. For $x = 0.2$, two magnetic phases appear upon cooling[27]. Neutron diffraction reveals ordered moments predominantly along the crystallographic $b$-axis below $T_2 = 25$ K, while below $T_1 = 21$ K, the moments partially reorient towards the $a$-axis in a low-temperature oblique phase. Our investigations of the field-induced polarization in these phases have uncovered a complex ME coupling scheme. The lowered magnetic symmetry of the oblique phase combined with the broken discrete translational symmetry, unlocks ME tensor elements that are otherwise forbidden in the parent compounds. Simulations show that the key factors responsible for the observed oblique phase are mismatched anisotropies combined with an inter-species exchange coupling creating competing exchange and single-ion anisotropy energy terms. This unusual mechanism is of general applicability and represents a promising approach to search for oblique ME phases in other families of compounds where the ME properties can be chemically tuned.

## Results

### Magnetic susceptibility

Figure 1c–e illustrate distinct differences in magnetic susceptibility between LiNi$_{0.8}$Fe$_{0.2}$PO$_4$ and its parent compounds, LiNiPO$_4$ and LiFePO$_4$. The susceptibility curves, $\chi_a$, $\chi_b$ and $\chi_c$, of both LiFePO$_4$ and LiNiPO$_4$ for fields along $a$, $b$ and $c$ display textbook behavior for antiferromagnets with easy axes along $b$ and $c$, respectively. The component of $\chi$ parallel to the easy axis drops towards zero below the transition temperature while the two perpendicular components remain nearly constant. By contrast, the susceptibility of LiNi$_{0.8}$Fe$_{0.2}$PO$_4$ shows clear evidence of two magnetic phase transitions. Below $T_2 = 25$ K, $\chi_b$ decreases while $\chi_a$ and $\chi_c$ remain constant. At a slightly lower temperature, $T_1 = 21$ K, $\chi_a$ begins to drop precipitously and the decrease of $\chi_b$ is interrupted, while $\chi_c$ remains approximately constant. These observations are indicative of a negligible $c$-axis component of the ordered moment at all temperatures, and of a rotation of the ordered moments from the $b$ axis towards the $a$ axis for temperatures lower than $T_1$. These two transitions were previously reported and we compare our findings with those of the authors of ref. 27 later in the Results section. Note that overall the susceptibility of the mixed system is higher than for the parent compounds. This, together with the overall different temperature dependence of the susceptibility as compared to the parent compounds, is evidence that LiNi$_{0.8}$Fe$_{0.2}$PO$_4$ is indeed a solid solution and we can exclude phase separation in the system.

## Magnetic structures

To determine the magnetic structures in LiNi$_{0.8}$Fe$_{0.2}$PO$_4$ we turn to neutron diffraction. At all temperatures below $T_2$, the commensurate magnetic Bragg peaks were found to be resolution limited, implying long-range order (Supplementary Fig. 2d–f in the Supplementary Information). A representative selection of temperature-dependent integrated intensities as obtained at the diffractometer, E5, is shown in Fig. 2a. The intensity of each magnetic Bragg peak reflects different combinations of symmetry components of the magnetic order. In addition, it carries information about the spin orientation in the ordered states, because neutrons couple exclusively to components of the magnetic moment perpendicular to the scattering vector $\mathbf{Q}$ (see Supplementary Table I). Our analysis indicates that the main magnetic structure component at all temperatures below $T_2$ is (↑↑↓↓) with the numbering of spins defined in Fig. 1a. Rietveld refinement of the magnetic Bragg peak intensities at base temperature yields magnetic moments predominantly in the $(a, b)$-plane with major component along $a$. For $T_1 \leq T \leq T_2$, our data suggests moments aligned along $b$.

The two transitions observed in our susceptibility measurements have clear signatures in the diffraction data: The $(0, 0, -1)$ and $(3, 0, -1)$ reflections grow linearly with decreasing temperature below $T_2 \approx 25$ K. By contrast, the $(0, 1, 0)$ peak appears only below $T_1 \approx 21$ K where in addition, there is a kink in the temperature profile of the $(3, 0, -1)$ intensity. The temperature dependencies of all recorded peaks are well described by a combination of a linear function and a power law, reflecting the existence of two order parameters, below $T_2$ and $T_1$, respectively (solid lines in Fig. 2a and in Supplementary Fig. 1). Simultaneous fits to all data sets yield transition temperatures $T_2 = 25.7(2)$ K and $T_1 = 20.8(1)$ K respectively, in good agreement with refs. 27 and 28. We note that the critical exponents for the two order parameters are clearly different. Below $T_2$, the neutron intensity increases linearly with decreasing temperature which means a critical exponent of $\frac{1}{2}$ as assumed fixed in the fit. This corresponds to the critical exponent resulting from long-range interactions or from a secondary order parameter. At $T_1$, $(0, 1, 0)$ displays a power law behavior with $\beta = 0.32(3)$ which is comparable to the critical exponent of a 3D Heisenberg, XY or Ising system.

To unambiguously determine the spin orientations, we performed a polarized neutron diffraction experiment using the triple axis spectrometer 4F1 and with scattering vector $\mathbf{Q} = (0, K, L)$ in the horizontal scattering plane. Uniaxial polarization analysis allows the two spin components perpendicular to $\mathbf{Q}$ to be individually addressed. This is done by measuring spin-flip (SF) and non spin-flip (NSF) intensities for the neutron beam polarization along the scattering vector ($P||x$), perpendicular to $\mathbf{Q}$ in the horizontal scattering plane ($P||y$), and along the direction perpendicular to the scattering plane ($P||z$). The temperature-dependencies of the resulting six cross sections were collected for the $(0, 1, 0)$, $(0, 0, 1)$ and $(0, 1, 2)$ reflections. The SF cross sections carry information on spin components perpendicular to both $\mathbf{Q}$ and the neutron beam polarization $\mathbf{P}$. The NSF cross sections reveal spin components perpendicular to $\mathbf{Q}$ but parallel to $\mathbf{P}$ in addition to any finite nuclear Bragg peak intensity.

Noting that the $(0, 1, 0)$ magnetic peak exclusively reflects (↑↑↓↓) symmetry components (Supplementary Table I), Fig. 2b, c show that the magnetic structure below $T_1$ involves sizeable spin components along $a$, but only negligible $c$-axis components. Spin components parallel to $b$ do not contribute to magnetic scattering at $\mathbf{Q} = (0, 1, 0)$, but can be probed at $\mathbf{Q} = (0, 0, 1)$ or $(0, 1, 2)$. Figure 2d, e confirm the involvement of an $a$-axis spin components below $T_1$, and show that the scattering is dominated by spins oriented along $b$ in the range $T_1 \leq T \leq T_2$. Note that here we plot only data for $(0, 1, 0)$ and $(0, 0, 1)$ as their interpretation is straightforward. The data for $(0, 1, 2)$ is shown in Supplementary Fig. 2a, b. A comparison of the observed intensities to the structure factors for the magnetic symmetry components contributing to the $(0, 1, 0)$, $(0, 0, 1)$ and $(0, 1, 2)$ peaks makes it clear

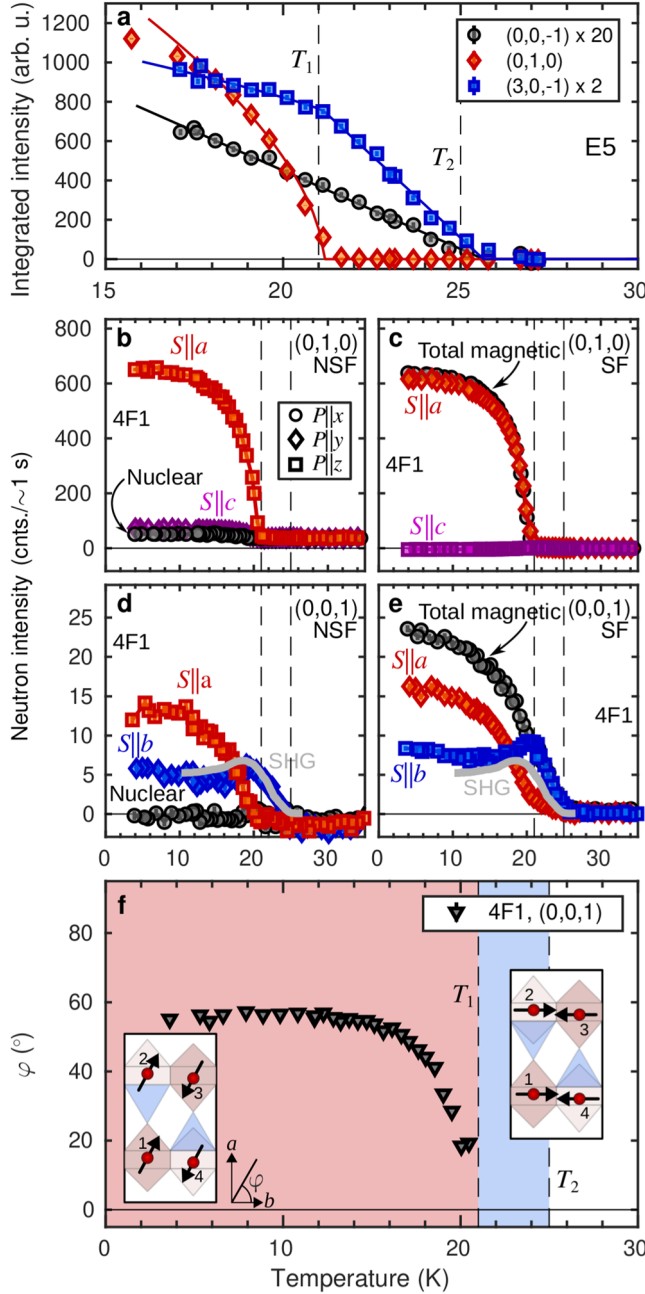

**Fig. 2 | Neutron diffraction. a** Temperature profiles of neutron diffraction intensities for three magnetic Bragg peaks for LiNi$_{0.8}$Fe$_{0.2}$PO$_4$. Circle, diamond and square symbols denote (0, 0, −1), (0, 1, 0) and (3, 0−1) respectively. Intensities are scaled as indicated in the legend. The solid lines show the fits described in the main text. Peak intensities obtained with polarized neutrons in the NSF (**b** and **d**) and SF (**c** and **e**) channels after correcting for non-perfect beam polarization[55]. Circle, diamond and square symbols are for neutron polarizations along x, y and z respectively. The symbols are coloured black for nuclear or total magnetic scattering, red for $S\|a$, blue for $S\|b$ and purple for $S\|c$. The crystal was oriented with the a axis vertical such that NSF intensities for $P\|z$ reflect a axis spin components for both (0,1,0) and (0,0,1). Grey curves in **d**, **e** show optical second harmonic generation measurements reprinted with permission from ref. 27. Copyright (2023) by the American Physical Society. **f** Moment rotation angle, φ, as a function of temperature with insets showing projections in the (0, 0, 1) plane of the magnetic structures for $T \leq T_1$ and $T_1 \leq T \leq T_2$. The red and blue shadings illustrate the extends of the two respective phases. The two transitions at $T_1$ and $T_2$ are marked by vertical dashed lines in all panels. Error bars of the neutron counts, N, follow Poisson statistics as $\sqrt{N}$ and the errors in **f** are propagated from the neutron counts. Error bars are smaller than symbol sizes in all panels.

that the dominant symmetry component for $T_1 \leq T \leq T_2$ is also (↑↑↓↓). The scattering from b-axis spin components, reflected by the NSF, $P\|y$ and SF, $P\|z$ cross sections in Fig. 2d, e increases monotonically for temperatures in the range $T_1 \leq T \leq T_2$ and levels off to a finite value at our experimental base temperature. The rotation angle, φ, in the (a, b)-plane may be calculated from the ratio of $P\|y$ and $P\|z$ data in Fig. 2d, e leading to the conclusion that the angle between the moments and the b axis approaches φ = 60° at low temperatures (Fig. 2f).

The small but finite nuclear intensity for $P\|x$ in Fig. 2b and Supplementary Fig. 2a may be due to a change of the lattice symmetry which could be caused by magnetostriction. Magnetostriction is common in magnetoelectrics and for LiFePO$_4$ this effect has been observed when applying magnetic fields[29]. Future synchrotron X-ray studies will uncover the evolution of the crystal lattice and symmetry as a function of temperature.

The solid grey lines in Fig. 2d, e represent the intensity of the second harmonic generation (SHG) susceptibility tensor element, $\chi_{zxx}$, from ref. 27. Here the first subscript signifies the component of the non-linear polarization induced by an electric field with components denoted by the last two subscripts. The similarity of the SHG signal with the NSF, $P\|y$ and SF, $P\|z$ cross sections is clear evidence that these two observations are intimately related. The SHG data was interpreted by the authors of ref. 27 as a signature of spin rotation from the easy b axis of stoichiometric LiFePO$_4$ towards the easy c axis of stoichiometric LiNiPO$_4$, upon cooling below $T_1$. Our polarized neutron diffraction results only allow for a small spin component along c and show instead a sizeable component along a. This picture is consistent with the susceptibility data in Fig. 1c. The physical mechanism for this surprising reorientation away from the easy axes of the two parent compounds is explored in our Monte Carlo simulations to be presented further on, but first we look into its profound consequences for the ME coupling.

## Magnetoelectric effect

The linear ME effect is described by the relation $\mathbf{P}^E = \alpha\mathbf{H}$ between the components of the induced electrical polarization, $\mathbf{P}^E$, and those of the applied magnetic field, $\mathbf{H}$. A related equation, $\mu_0\mathbf{M} = \alpha^T\mathbf{E}$, connects the components of the induced magnetization, $\mathbf{M}$, to those of the applied electric field, $\mathbf{E}$. For systems invariant to integer lattice vector translations, the allowed elements of the ME tensor $\alpha$ are imposed by the point group symmetry of the magnetically ordered state[14,15]. Specifically, for the stoichiometric parent compounds LiNiPO$_4$ and LiFePO$_4$, the reported magnetic structures imply that the elements which may be non-zero are $\alpha_{ac}$, $\alpha_{ca}$ and $\alpha_{ab}$, $\alpha_{ba}$, respectively.

The ME response of LiNi$_{0.8}$Fe$_{0.2}$PO$_4$ was probed with measurements of the pyrocurrent produced by a temperature change (see Methods and Supplementary Information for details). Our results for LiNi$_{0.8}$Fe$_{0.2}$PO$_4$ are shown in Fig. 3 and are compared to the ME response of the parent compounds, LiNiPO$_4$[19] and LiFePO$_4$[30]. Note that in the following analysis we assume space group Pnma although it was recently shown that LiFePO$_4$ may display a lower symmetry[30]. The pyrocurrent for LiNi$_{0.8}$Fe$_{0.2}$PO$_4$ for two orthogonal orientations of the electric poling field, $\mathbf{E}$, and three directions of the magnetic field shows clear signatures of two ME phase transitions slightly below $T_2$ and $T_1$, see Fig. 3a–c. The evidence is in the form of spikes in the pyrocurrent, which following a geometrical correction can be integrated to obtain the temperature dependent polarization components, $P_i^E$. A signal is thus observed for all probed couplings except $\alpha_{bc}$.

The electric polarization corresponding to the tensor elements $\alpha_{ab}$ and $\alpha_{ba}$ together with that corresponding to $\alpha_{ac}$ are shown in Fig. 3d. As mentioned above, these components are known to be non-zero for stoichiometric LiFePO$_4$ and LiNiPO$_4$[23,31]. When comparing the ME response of LiNi$_{0.8}$Fe$_{0.2}$PO$_4$ to that of LiFePO$_4$ and LiNiPO$_4$ measured under identical conditions (blue and grey dashed lines in Fig. 3d), it is apparent that the polarizations induced along a and b are

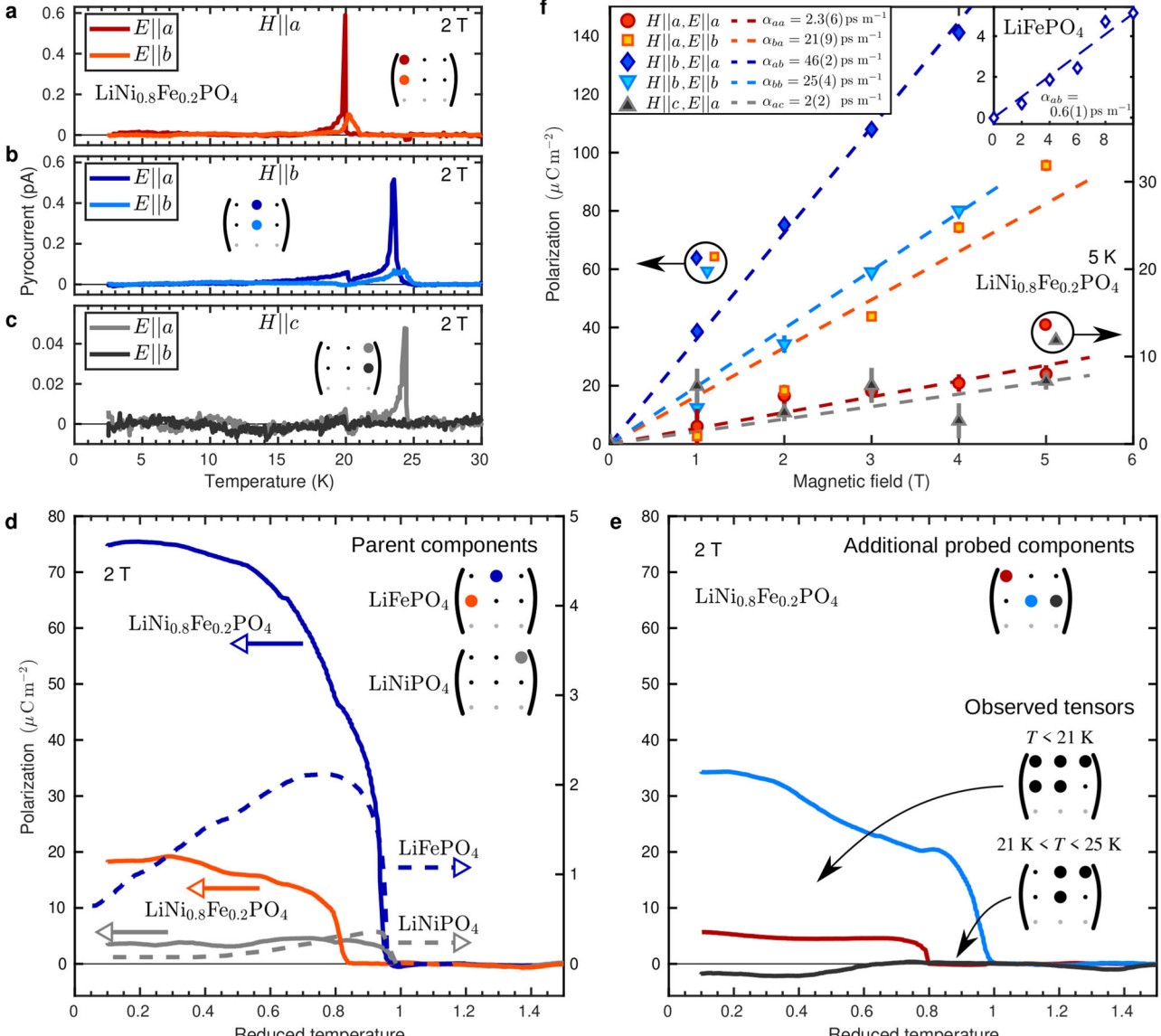

**Fig. 3 | Pyrocurrent and magnetoelectric effect.** Panels **a-c** show the pyrocurrent for LiNi$_{0.8}$Fe$_{0.2}$PO$_4$ as a function of temperature for magnetic fields applied along $a$ (red and orange curves), $b$ (dark and light blue curves) and $c$ (grey and black curves), respectively. The insets indicate which elements of the ME tensor, $\alpha$, were probed. The error on the measured current is of the order of 5 fA. The colour codes given in **a-c** are followed in the remaining panels of this figure. **d** Electric polarization as a function the reduced temperature with transition temperatures 21, 25 and 50 K at zero field for LiNiPO$_4$, LiNi$_{0.8}$Fe$_{0.2}$PO$_4$ and LiFePO$_4$, respectively. Note the two $y$-axes: the left one for the data for the mixed system (solid lines) and the right one for the parent compounds (dashed lines). The curve shown for LiNiPO$_4$ is from ref. 19 and that for LiFePO$_4$ is reprinted with permission from ref. 30. Copyright (2023) by the American Physical Society. **e** Temperature dependency of the electric polarization originating from tensor elements not present in the parent compounds. For $T < T_1$ and $T_1 < T < T_2$, all observed non-zero ME tensor elements in LiNi$_{0.8}$Fe$_{0.2}$PO$_4$ are indicated. The measurements shown in **a-e** were carried out with an applied magnetic field strength of 2 T where the ME effect is still linear. The errors on the polarization are of the order of $1\,\mu$Cm$^{-2}$. **f** Field dependency of the average of the induced electric polarization for $T < 5$ K for non-zero couplings. The error bars are estimated from the variations observed in the temperature profiles of the polarization (see Supplementary Fig. 4). Note that strong and weak ME components are plotted on two different $y$-axes as illustrated with encircled symbols and arrows. The dashed lines are linear fits, $P_i^E = \alpha_{ij}H_j$, to the data with the obtained ME coefficients, $\alpha_{ij}$, listed in the legend. The inset shows the corresponding data for LiFePO$_4$ with $H\|b, E\|a$.

significantly larger in LiNi$_{0.8}$Fe$_{0.2}$PO$_4$ at all temperatures below the transition temperature. Most strikingly, in the limit $T \to 0$, the polarization due to the dominant tensor component, $\alpha_{ab}$ is increased by almost two orders of magnitude compared to LiFePO$_4$. A second remarkable observation is that the onset temperatures of $\alpha_{ab}$ and $\alpha_{ba}$ are different. $\alpha_{ba}$ vanishes in the range $T_1 \leq T \leq T_2$ whereas $\alpha_{ab}$ is finite already below $T_2$ and displays a kink at $T_1$.

Finally, in Fig. 3e we probe tensor components that are by symmetry not allowed for LiNiPO$_4$ and very small for LiFePO$_4$[30]. Similarly, we observe $\alpha_{bb}$ below $T_2$ while $\alpha_{aa}$ is finite only below $T_1$. For the last tensor

element measured, $\alpha_{bc}$, there is no spike to be seen in the pyrocurrent and we conclude that this element is either very weak or zero.

By measuring the pyrocurrent at different magnetic field strengths, we obtain the electric polarization as a function of field as shown in Fig. 3f. The values of the polarization shown here are the mean values for temperatures below 5 K. For the corresponding polarization curves at the different field strengths, see Supplementary Fig. 4. The measured polarization is linear with field for most couplings, except notably for $\alpha_{ba}$. Interestingly, $\alpha_{ba}$ is exactly the component with a different onset temperature compared to $\alpha_{ab}$, underlining that the behavior of the ME

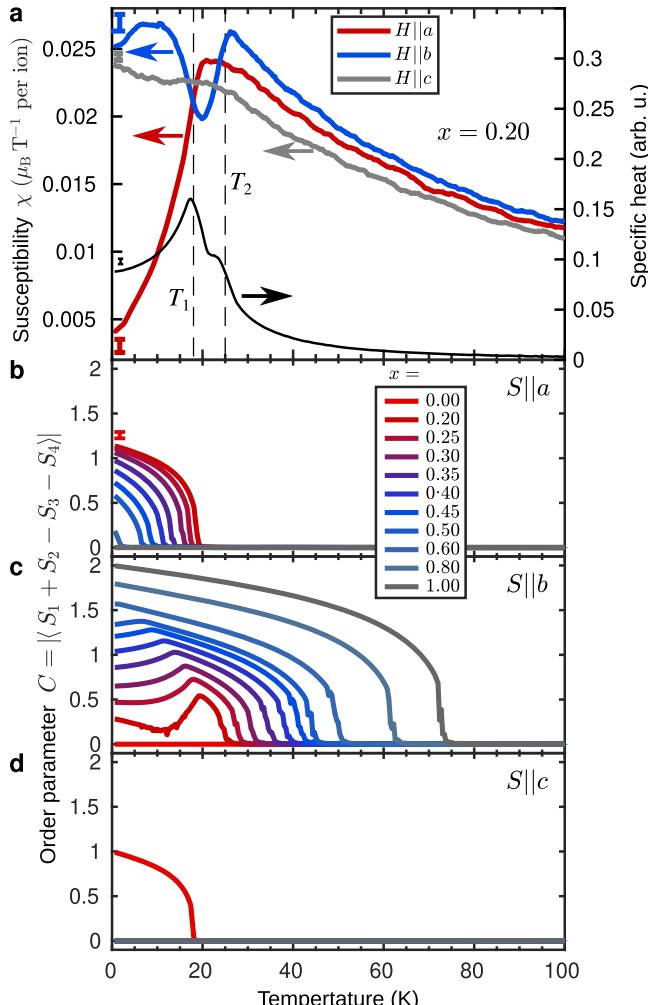

**Fig. 4 | Monte Carlo simulations. a** Magnetic susceptibility (left axis) for fields along $a$ (red curve), $b$ (blue curve) and $c$ (grey curve) and specific heat (right axis, black curve) for LiNi$_{1-x}$Fe$_x$PO$_4$ with $x = 0.20$. **b–d** The absolute value of the $C$-type order parameter plotted as a function of temperature for spin components along the three crystallographic axes and for different values of $x$. The colours of the curves illustrate the Fe concentration from red ($x = 0$) over blue to grey ($x = 1$). Note that in **d** the curves for $x \geq 0.20$ coincide. The maximum error bar sizes for each of the variables are indicated at the lowest temperatures in **a** and **b**.

effect in LiNi$_{1-x}$Fe$_x$PO$_4$ is complex. Nevertheless, it is clear from our measurements that the effect is much stronger in the mixed system compared to the parent compounds and that at low field strengths, the system is in the linear regime.

**Monte Carlo simulations.** We now show that classical Monte Carlo simulations reproduce the salient features of the susceptibility and diffraction results for $x = 0.20$. In the calculations, we chose $J_{bc} = 1$ meV and $J_{ab} = 0.3$ meV for all corresponding pairs of sites (see Fig. 1a), irrespective of their occupancy by Ni or Fe. For the single-ion anisotropies we used $(D^a, D^b, D^c) = (0.3, 1.8, 0)$ meV for Ni with $S = 1$ and $(D^a, D^b, D^c) = (0.6, 0, 1.6)$ meV for Fe with $S = 2$. This choice of parameters looks like a dramatic simplification when compared to the full set of experimentally determined parameters[25,26,32] for LiNiPO$_4$ and LiFePO$_4$ as given in Supplementary Table IV. Nevertheless, the set of simplified parameters yields results in good agreement with our experimental neutron diffraction and susceptibility measurements for $x = 0.20$. In turn, this agreement justifies the use of the Hamiltonian from Equation (1) with the chosen parameters for an exploration of the full phase diagram of LiNi$_{1-x}$Fe$_x$PO$_4$.

In Fig. 4a, we plot the calculated magnetic susceptibility and specific heat for $x = 0.20$ as a function of temperature. Two phase transitions are observed near 25 and 20 K. The transition temperatures as well as the temperature-dependencies of the three components of $\chi$ are in excellent agreement with the experimental results shown in Fig. 1c. The accuracy of the simulations is further illustrated by comparing the calculated and measured susceptibilities for $x = 0.06$ with the corresponding experimental data (Supplementary Fig. 5).

Next, we study the simulated $C$-type order parameter $C = |\langle S_1 + S_2 - S_3 - S_4 \rangle|$ for spin components along $a$, $b$ and $c$, respectively. Figure 4b–d show the temperature dependencies of $C$ for the full range of Fe concentrations, $x$. Focusing on $x = 0.20$, the resemblance with the polarized neutron diffraction data in Fig. 2 is striking. Note that the neutron diffraction intensity is proportional to the moment squared. The first phase transition at 25 K corresponds to spins ordering along $b$. The growth of the corresponding order parameters is interrupted at 20 K where a rotation towards $a$ starts and the oblique low-temperature phase is entered. The $c$-axis component remains zero at all temperatures. From the $a$ and $b$ components of the simulated order parameter we arrive at a rotation angle of 76° at low temperature, which compares reasonably well with the value of ≈60° obtained from the experimental data (Fig. 2f). We use the transition temperatures derived from the simulated specific heat and order parameters data to construct the $(x, T)$ phase diagram shown in Fig. 1b. The simulations underestimate the transition temperatures for small $x$ compared to the measured values (star symbols for $x = 0$, 0.06 and 0.20), but the ratios of simulated to observed transition temperatures are relatively constant with $x$ in this range.

The oblique antiferromagnetic state is relatively robust. The simulations show that the only requirements are an inter-species exchange interaction as well as competing single-ion anisotropies with opposite easy and hard axes for the parent compounds but a common intermediate axis off any of these easy and hard axes. It is this frustration between exchange and single-ion anisotropy energies that generates the oblique state. In the analysis of the neutron diffraction data we assumed a collective behavior of all spins regardless of species. The simulations show that indeed the ensemble average of the moments give a collectively ordered picture. However, we also find local fluctuations between Ni and Fe sites (Supplementary Fig. 6). The ordered moment for the oblique phase is therefore lower when calculating the average over the entire system than when considering individual sites, not only due to thermal fluctuations but also due to site specific differences in the moment orientation. This effect and the general consequences of violation of discrete translational symmetry for the ME effect are an interesting topic of future theoretical investigations.

## Discussion

The effects on the magnetic ground state of a quenched random distribution of ions in mixed anisotropy antiferromagnets have been extensively studied by renormalization group theory and in various mean field models[16–18,33–36]. The resulting phase diagrams generically contain one or more oblique phases at intermediate compositions, in which the ordered moments are oriented away from the easy axes of the parent compounds. Depending on the details of the exchange and anisotropy terms in the spin Hamiltonian, the oblique ground state may involve ordered moments in a plane spanned by the easy axes of the two parent compounds, or perpendicular to both in the particular case where the easy and hard axis of one species coincide with the hard and easy axes of the other species[17]. These predictions of a chemically tunable magnetic ground state are broadly consistent with our experimental observations and Monte Carlo simulations for LiNi$_{1-x}$Fe$_x$PO$_4$, and have in the past been found to agree well with experimental studies of other mixed-anisotropy antiferromagnets[37–42].

In the lithium orthophosphates, the magnetic $C$-type structure is the dominant structure component for all stoichiometric family

members. The single-ion anisotropy plays a crucial role for the ME coupling, as the allowed tensor elements derive from the magnetic point group, ipso facto the direction of the ordered moments. Since $S||b$ in LiFePO$_4$, $\alpha_{ab}$ and $\alpha_{ba}$ are allowed, while as $S||c$ in LiNiPO$_4$, $\alpha_{ac}$ and $\alpha_{ca}$ are allowed. When $S||a$, the diagonal elements are allowed ($\alpha_{aa}$, $\alpha_{bb}$, $\alpha_{cc} \neq 0$). It follows from our results, however, that such simple rules do not apply in the mixed systems, where discrete translational symmetry is broken and the local spin anisotropy is site dependent. Between 21 and 25 K, the predominant spin orientation in LiNi$_{0.8}$Fe$_{0.2}$PO$_4$ is $S||b$. Nevertheless, the observed diagonal tensor element $\alpha_{bb}$ is almost as strong as the expected $\alpha_{ab}$-component, while $\alpha_{ba} = 0$. Below 21 K, the ME-tensor is more complex and most tensor elements are observed. This is due to the off-axis direction of the ordered moments. However, the fact that some expected tensor elements are extinct between 21 and 25 K, while some unexpected elements are not, is a strong hint that while discrete translational symmetry is broken, the existence of the ME coupling is still governed by magnetic point group symmetry, yet not in the same manner as in the stochiometric systems. This is an interesting point in itself and should be subject to further theoretical study.

The most intriguing observation is the almost hundredfold increase in the strength of the ME coupling for LiNi$_{0.8}$Fe$_{0.2}$PO$_4$ observed in the pyrocurrent measurements. In LiNiPO$_4$ and LiFePO$_4$, the effect is believed to arise from exchange striction[22,26,43,44]. Specifically, a microscopic model connects the electric polarization caused by a displacement, $x_i$, of PO$_4$ tetrahedra along $i$, to the component of the applied magnetic field along $j$ as follows:

$$P_i^E \propto x_i = \frac{\lambda_i}{\epsilon_i} \langle S \rangle^2 \chi_j H_j. \qquad (2)$$

Here $\langle S \rangle$ is the order parameter, $\chi_j$ the magnetic susceptibility for fields along $j$, $\epsilon_i$ the elastic energy constant associated with tetrahedron displacement ($\mathcal{E}_i = \epsilon_i x_i^2$) and $\lambda_i$ reflects the strength of the exchange striction ($J_{H \neq 0} \to J_{H=0} + \lambda_i x_i$). In addition to the general increase in magnetic suceptibility in LiNi$_{0.8}$Fe$_{0.2}$PO$_4$ as compared to the parents, both a reduction of the elastic displacement energy and an increase in exchange striction could cause stronger ME couplings. A lowering of crystal symmetry may indeed result in lower energy cost for displacing the PO$_4$ tetrahedra as well as more allowed options for displacement directions, i.e. a softer lattice. Moreover, the local number of nearest neighbors of a given species, and variations thereof, could bring exchange striction terms into play which would otherwise cancel out (in the parent compounds only interactions between ion pairs (1,2) and (3,4) contribute to the ME effect[22]).

Tuning of the magnetic symmetry was also recently achieved in the olivine series of compounds, Li$_{1-x}$Fe$_x$Mn$_{1-x}$PO$_4$[45], as well as in Mn and Co doped LiNiPO$_4$[46]. These studies do not report on the corresponding consequences for the ME effect but they further illustrate that the lithium orthophosphate family harbour many possibilities for tailoring the magnetic and consequently also ME material properties. In general, the existence of ME and multiferroic oblique antiferromagnets is unlikely to be limited to this family. Notably, our Monte Carlo simulations show that competing single-ion anisotropies with a common intermediate axis combined with a significant inter-species exchange coupling are the decisive ingredients to prodcue an oblique magnetoelectric state, whereas details of the exchange coupling scheme play almost no role. Generally, transition metal ions exhibit complex single-ion anisotropies in octahedral environments, and will likely share an intermediate anisotropy axis in many families of compounds. An oblique ME phase may therefore also exist in other classes of materials. In future studies, Monte Carlo simulations similar to those performed in this work, possibly combined with DFT calculations to determine exchange constants, could precede time-demanding material synthesis in order to predict the viability of the candidate family to produce oblique ME phases. An interesting family of

compounds for future studies of this type is for example Mn$_{1-x}$M$_x$WO$_4$ ($M$ = Fe, Co, Cu, Zn)[47–50].

In summary, we studied LiNi$_{0.8}$Fe$_{0.2}$PO$_4$ experimentally using magnetometry, neutron diffraction and pyrocurrent measurements and theoretically through Monte Carlo simulations. We have identified an oblique low-temperature phase over an extended range of compositions. In this phase the spins rotate away from the distinct easy axes of the parent compounds, LiNiPO$_4$ and LiFePO$_4$, towards the direction of the common intermediate axis. The magnetoelectric properties correlate with the observed magnetic phase transitions, but the form of the magnetoelectric tensor departs from theoretical expectations based on discrete translational invariance and magnetic point group symmetries. Most dramatically, we observe a strong enhancement of almost two orders of magnitude for the dominant magnetoelectric tensor element compared to the parent compounds. These data in combination with our Monte Carlo simulation results suggest that the observations have broader implications and that chemical tuning of oblique magnetoelectric phases represents a promising path for tailoring magnetoelectric material properties.

## Methods

### Magnetic susceptibility
Magnetization measurements were performed using a Cryogenic Ltd. cryogen free vibrating sample magnetometer. A magnetic field of 0.5 T was applied along $a$ and $b$ for temperatures in the range 2–300 K. For measurements with field along $c$ we used a Physical Property Measurement System (PPMS) from Quantum Design. The sample for all measurements of LiNi$_{0.8}$Fe$_{0.2}$PO$_4$ was a 25 mg box-shaped single crystal.

### Neutron diffraction
Unpolarized neutron diffraction data on LiNi$_{0.8}$Fe$_{0.2}$PO$_4$ were collected at the E5 diffractometer at the Helmholtz-Zentrum Berlin using an 4-circle cradle, a neutron wavelength of $\lambda = 2.38$ Å selected by a pyrolytic graphite (PG) monochromator and a 2D position sensitive neutron detector of size 9 × 9 cm$^2$. A PG filter before the sample position was used for suppression of second order neutrons ($\lambda/2$) from the monochromator.

The polarized neutron diffraction data were obtained on the 4F1 triple-axis spectrometer at the Laboratoire Léon Brillouin, using a wavelength of $\lambda = 2.44$ Å. The monochromatic incident beam was polarized using a bending mirror and a PG filter after the bender reduced second order contamination of the incident beam. A spin flipper was placed before the sample position. In combination with a Heusler analyzer this allowed for probing both spin flip and non spin-flip scattering. A set of Helmholtz-like coils around the sample position enabled polarization of the neutron beam along $x$, $y$ or $z$ in the coordinate system decribed in the Supplementary Information. The same high-quality 250 mg LiNi$_{0.8}$Fe$_{0.2}$PO$_4$ single crystal was used in both polarized and unpolarized neutron diffraction experiments. At 4F1, it was oriented with (0, $K$, $L$) in the horizontal scattering plane. Flipping ratios $F = 40$ and 25 were deduced from measurements on the structural (020) and (002) reflections, and used to correct for non-ideal beam polarization.

Preliminary studies of the magnetic structures of LiNi$_{1-x}$Fe$_x$PO$_4$ were carried out at the triple-axis spectrometer RITA-II and diffractometer TriCS at the Paul Scherrer Institute.

### Pyrocurrent measurements
The quasi-static method[51] was used to perform pyrocurrent measurements with a Quantum Design PPMS at the Helmholtz-Zentrum Berlin. The custom-build insert allows for sample rotations with respect to the vertical magnetic field and thus enables probing off-diagonal ME tensor elements[30,52]. Plate-shaped LiNi$_{0.8}$Fe$_{0.2}$PO$_4$ single crystals with gold sputtered faces perpendicular to $a$ and $b$, and sample thickness 0.5 and 0.9 mm, respectively, were used. Single crystals of LiNiPO$_4$ and LiFePO$_4$

were similarly prepared with faces perpendicular to $a$ and thickness 0.6 and 0.5 mm, respectively. The precision of the cuts were within 0.5° which determines the uncertainty of the field direction with the respect to the axes perpendicular to the surface. For the other field directions, the crystals were aligned by eye to obtain a sample alignment within 2°. A potential of 100 V was applied as well as a magnetic field while cooling to obtain a single ferroelectric domain. The potential was switched off at the experimental base temperature. The measurement was then performed upon heating at a constant temperature rate. Magnetic fields were applied along the $a$, $b$ and $c$ directions.

## Monte Carlo simulations

Classical Monte Carlo simulations were carried out using the spin Hamiltonian [Equation (1)] and employing the Metropolis algorithm[53]. The system was of size $10 \times 10 \times 10$ crystallographic unit cells (i.e. 4000 magnetic sites) and with Fe and Ni ions randomly distributed. The simulation was run for $10^5$ Monte Carlo steps for each temperature in the range 1–100 K with step size 0.35 K and decreasing temperature. For each temperature we use the final configuration from the previous temperature step as a starting point. This procedure mimics the process of slowly lowering the temperature in the physical experiment. All simulations were run at zero field.

For each value of the Fe concentration, $x$, we simulated 5 distinct configurations from which we calculated the magnetic susceptibility, specific heat and order parameter. The susceptibility is calculated as $\chi = \beta(\langle M^2 \rangle - \langle M \rangle^2)$, where $M$ denotes the total magnetization of the system, $\beta = \frac{1}{k_B T}$ and $k_B$ is the Boltzmann constant. The brackets, $\langle \rangle$, denote the ensemble average over system configurations. In Fig. 4a and in Supplementary Fig. 5b we show $\chi$ normalized per magnetic site. The specific heat is calculated from the energy dissipation theorem, $C_V = k_B \beta^2 (\langle E^2 \rangle - \langle E \rangle^2)$, where $E$ is the total energy of the system. The order parameter is calculated as an average over all unit cells, each containing four magnetic sites, i.e. $C = |\langle S_1 + S_2 - S_3 - S_4 \rangle|$. The curves shown in Fig. 4b–d are then the average quantities over the 5 configurations.

## Data availability

The data that support the findings of this study are available at https://doi.org/10.5281/zenodo.7515107 and from E.F. upon request. Source data are provided with this paper.

## Code availability

The Monte Carlo code used in this study is available from E.F. upon reasonable request.

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

## Acknowledgements

This work was supported by the European Research Council through the Synergy network HERO (Grant No. 810451). We thank the EU Interreg program MAXESS4FUN for Cross Border and Society for funding the simulation work. We are grateful for neutron beamtime received for this project at the instruments TriCS and RITA-II at the SINQ neutron spal-lation source at the Paul Scherrer Institute, at the E5 diffractometer at the BER-II research reactor at the Helmholtz-Zentrum Berlin, and at the 4F1 spectrometer at the research reactor at the Laboratoire Leon Brillouin. We acknowledge Diamond Light Source for beamtime on I16 (Proposal No. MM30817-1). This project was supported by the Danish national Council for Research infrastructure (NUFI) through DANSCATT and the ESS-Lighthouse Q-MAT. Ames National Laboratory is operated for the U.S. Department of Energy by Iowa State University under Contract No. DEAC02-07CH11358. We thank M. Laver for assistance with neutron scattering experiments, A. Sokolowski for support with pyrocurrent measurements and J. Li for samples.

## Author contributions

The experimental project and theoretical framework was conceived by E.F., R.T.-P. and N.B.C. The crystals were grown by D.V. Magnetic sus-ceptibility measurements were performed by E.F., K.S.P. and K.K.M. Diffraction experiments were performed by E.F., R.E.H., A.B.K., M.K.S., G.M., A.B., J.-R.S. M.R., P.B., S.H.-D., O.Z., J.S., Ch.N. R.T.-P. and N.B.C. Pyrocurrent measurements were carried out by E.F., B.K., S.P. and R.T.-P. Sample preparation for the pyrocurrent measurements was performed by E.F. and A.P. E.F. implemented and performed Monte Carlo simula-tions with support from O.F.S. Data analysis and figure preparation were performed by E.F. Interpretation and manuscript writing was done by E.F., R.T.-P., N.B.C. and H.M.R. with contributions from all authors.

## Competing interests

The authors declare no competing interests.
