## [Peer Review File · Nature Communications]

Reviewers' Comments:

Reviewer #1:

Remarks to the Author:

The authors studied chemical tuning in mixed-anisotropy antiferromagnets to tune the properties of single-phase magnetoelectric $\text{Li}(\text{Ni,Fe})\text{PO}_4$ using magnetometry, neutron diffraction, pyrocurrent measurements, and Monte Carlo simulations. In the $\text{LiNi}_{0.8}\text{Fe}_{0.2}\text{PO}_4$ phase, the spins rotate away from the distinct easy axes of the parent compounds, LiNiPO_4 and LiFePO_4 , towards the direction of the common intermediate axis. The authors studied the magnetoelectric properties by pyrocurrent measurements at a fixed magnetic field. This is a nice piece of work. However, the way the linear magnetoelectric coefficient is extracted based on a fixed DC magnetic field is questionable. An AC response under an AC magnetic or electric field excitation is needed to characterize the magnetoelectric properties. At a minimum, pyrocurrent measurements at different DC fields are needed to show the polarization change at different bias magnetic fields. I would suggest that the authors redo the pyrocurrent measurements at different bias magnetic fields for different compositions, and redo the evaluation of the magnetoelectric coupling coefficients.

Reviewer #2:

Remarks to the Author:

This is a very nice paper on a careful study of the magnetoelectric effect in the LiFePO_4 family of materials. This study uses a combination of multiple measurements and theory to illustrate the ME effect. The work is well thought out and the data looks solid. It would have been nice to have the converse ME effect data (i.e., change the magnetism with an electric field), but I think the use of crystals presumably precludes this (due to the large voltages required).

I would recommend publication of this nice piece of research.

Lausanne, 3 Feb 2023

Letter to Reviewers and List of Changes

We are grateful to the reviewers for their comments which have allowed us to further improve our manuscript NCOMMS-22-34598 with the title "*Tuning magnetoelectricity in a mixed-anisotropy antiferromagnet*". Please find below a list addressing the questions that were raised as well as a number of additional minor changes.

Reviewer #1

« The authors studied chemical tuning in mixed-anisotropy antiferromagnets to tune the properties of single-phase magnetoelectric $\text{Li}(\text{Ni},\text{Fe})\text{PO}_4$ using magnetometry, neutron diffraction, pyrocurrent measurements, and Monte Carlo simulations. In the $\text{LiNi}_{0.8}\text{Fe}_{0.2}\text{PO}_4$ phase, the spins rotate away from the distinct easy axes of the parent compounds, LiNiPO_4 and LiFePO_4 , towards the direction of the common intermediate axis. The authors studied the magnetoelectric properties by pyrocurrent measurements at a fixed magnetic field. This is a nice piece of work. However, the way the linear magnetoelectric coefficient is extracted based on a fixed DC magnetic field is questionable. An AC response under an AC magnetic or electric field excitation is needed to characterize the magnetoelectric properties. At a minimum, pyrocurrent measurements at different DC fields are needed to show the polarization change at different bias magnetic fields. I would suggest that the authors redo the pyrocurrent measurements at different bias magnetic fields for different compositions, and redo the evaluation of the magnetoelectric coupling coefficients. »

We thank reviewer 1 for the overall positive evaluation of our manuscript. Moreover, their questions made us realize that the manner in which we presented the analysis of the magnetoelectric (ME) coupling coefficients looked oversimplified because we only showed data obtained with a single applied magnetic field strength. As the reviewer points out, a more robust protocol would involve either an AC electric response to an AC magnetic field or data obtained with a range of different DC field strengths. Our original measurements were in fact of the latter kind and in hindsight this clearly should have been reflected in the origin manuscript. We have therefore carefully revisited the analysis including data in a field range up to 6 T and the field-dependent electric polarization has been introduced in Fig. 3f of the revised manuscript. For fields along **b** and **a**, field-induced phase transitions occur around 4 and 6 T respectively (see E. Fogh, PhD thesis, Technical University of Denmark, 2019). The understanding of these transitions is outside the scope of the present manuscript, which focuses on correlating magnetic structures with ME properties in the low-field phase. The complete set of temperature scans of the electric polarization at various DC magnetic fields

are shown in Fig. S4 of the Supplementary Information. The analysis of the extended data demonstrates linear dependencies of the magnetic-field-induced electric polarization for all ME couplings, except α_{ba} which displays a more complex scaling with field. These measurements clearly confirm the magnitude of the ME coupling strengths in our system. The enhancements of the strongest coefficient at low temperature is around 80 as compared to the parent compounds, i.e. not quite but almost the two orders of magnitude stated in the original manuscript. We have altered the formulation throughout the manuscript accordingly. The new panel in Fig.3 is accompanied by the following paragraph inserted at the end of the section about the ME effect :

« By measuring the pyrocurrent at different magnetic field strengths, we obtain the electric polarization as a function of field as shown in Fig. 3f. The values of the polarization shown here are the mean values for temperatures below 5 K. For the corresponding polarization curves at the different field strengths, see Fig. S4 in the Supplementary Information. The measured polarization is linear with field for most couplings, except notably for α_{ba} . Interestingly, α_{ba} is exactly the component with a different onset temperature compared to α_{ab} , underlining that the behavior of the ME effect in $\text{LiNi}_{1-x}\text{Fe}_x\text{PO}_4$ is complex. Nevertheless, it is clear from our measurements that the effect is much stronger in the mixed system compared to the parent compounds and that at low field strengths, the system is in the linear regime. »

The obtained ME coefficients are listed both in the legend of Fig.3f and in Table III of the Supplementary Information. We further added the following accompanying text in the Supplementary Information:

« Figure S4 shows the electric polarization as obtained from Eq.(3) for different magnetic field values. For $H||a$ and $H||b$, the ME curves are well defined but for $H||c$, the variations in the curves manifest a larger relative uncertainty. The anomalous behavior of the 1 T data is not understood. Linear fits to the average values of the polarization for $T < 5$ K (see Fig. 3f) yield the ME coefficients listed in Table III. For completeness, the result for α_{bc} is included although here the pyrocurrent exhibits no spike as a function of temperature.»

Having analyzed data with different magnetic bias fields, measuring an AC electric or magnetic response is not necessary. Our measurements have been performed on a general purpose PPMS cryomagnet using a compact custom sample holder. Unfortunately it is impossible to measure the ME effect using AC electric or magnetic fields with this setup. However, such measurements would indeed be interesting and should be part of future work.

The focus of the present manuscript is on understanding the magnetic and ME properties of the oblique phase in $\text{LiNi}_{1-x}\text{Fe}_x\text{PO}_4$. This phase was observed experimentally for $x = 0.20$ but not for $x = 0.06$ [29,30 and E. Fogh PhD thesis] nor in the parent compounds, i.e. $x = 0$ and 1. The (x,T) phase diagram as determined with Monte Carlo simulations shows good agreement with these observations. Powder synthesis for different compositions is the subject of a recently started PhD project but single crystals are needed for a full study of the ME tensor. Identifying the most interesting compositions from the powder samples and subsequently growing single crystals is a technically demanding process which takes several years and will thus not be part of the current manuscript.

Reviewer #2

« This is a very nice paper on a careful study of the magnetoelectric effect in the LiFePO₄ family of materials. This study uses a combination of multiple measurements and theory to illustrate the ME effect. The work is well thought out and the data looks solid. It would have been nice to have the converse ME effect data (i.e., change the magnetism with an electric field), but I think the use of crystals presumably precludes this (due to the large voltages required).

I would recommend publication of this nice piece of research. »

We thank reviewer 2 for their very positive feedback and the recommendation to publish our work. We agree that it would indeed have been nice to measure the magnetic response to an applied electric field to complement our study of the magnetic-field-induced electric polarization. In order to induce a measurable magnetization in the PPMS system, several tens of kV/mm should be applied, see e.g. Kocsis et al., *Phys. Rev. B* **104**, 054426 (2021). This is not possible in the existing setup and is in general very challenging. Nevertheless it is a good point and should be part of future investigations.

Other changes

After carefully reading through the manuscript again, we have introduced a number of minor changes as listed below. These merely serve to clarify the presentation and the conclusions are unaltered. Line numbers refer to the revised manuscript.

Ref. 34 has been published and the reference entry updated correspondingly :

[34] Fogh, E. et. al. The magnetoelectric effect in LiFePO₄ – revisited. *Physica B* **648**, 414390 (2022)

In the abstract we added a plural s in ‘applications’ line 31-32. We changed ‘eplacing a fracting’ with ‘partial substitution’ in line 35 to remove one word and achieve a total word count of 150 in the abstract.

In the paragraph commencing at line 105, we included ‘pyrocurrent’ such that it now reads :
« We have employed magnetic susceptibility and pyrocurrent measurements, neutron diffraction and Monte Carlo simulations to investigate the (x,T) phase diagram of LiNi_{1-x}Fe_xPO₄ (see Fig. 1b) »

In the very last sentence in the section about our neutron results in line 280, we replace ‘explore’ with ‘look into’ to avoid repetitions. The last part of the sentence now reads : « ...but first we look into its profound consequences for the ME coupling. »

In line 303, we replace ‘however’ with ‘although’ and merge the two sentences in lines 302-304 such the single sentence now reads « Note that in the following analysis we assume space group Pnma although it was recently shown that LiFePO₄ may display a lower symmetry ».

To clarify that there is no ME signal observe for E||b, H||c, we added the following sentence in line 312: « A signal is thus observed for all probed couplings except α_{bc} . »

As mentioned in the reply to reviewer 1, we added a new panel in Fig. 3. The layout of this figure has been modified accordingly to show all panels clearly. Additionally, in Fig. 3e, we added a black circle to indicate the ME tensor component, α_{bc} , under 'Additional probed components'. It was a mistake that it was not included before. The two topmost tensors shown in the previous panel f have been moved to panel e with arrows indicating the corresponding temperature intervals. The new panel f shows the field dependency of the electric polarization and the magnitude of the linear ME tensor components are now listed in this panel. We therefore remove the last of the tensors which in the previous panel f illustrated the relative strengths of these components.

In the caption of Fig. 3 we point out that the data shown for LiNiPO_4 and LiFePO_4 are reproduced from Refs. [22] and [34], respectively. It was not stated clearly in the original submission.

At the end of the section about the simulations, we replace 'issue' with 'effect' such that the sentence starting line 425 now reads : « *This effect and the general consequences of violation of discrete translational symmetry for the ME effect are an interesting topic of future theoretical investigations.* »

In line 497, we added « *..., i.e. a softer lattice* » to further explain our thoughts on why the ME effect is stronger in the mixed system as compared to the parent compounds.

In the Supplementary Information line 922-923 we corrected the coordinates of ions nr. 3 and 4 where the y-coordinate should be 3/4 rather than the previous 1/4.

All figure captions have been updated with information about error bars.

We are in the process of uploading all supporting data and analysis scripts to the sharing platform Zenodo. The repository will be published upon acceptance. We included source data in the resubmission. The data availability statement in the manuscript has been updated accordingly and now reads :

« *The data that support the findings of this study are available at <https://doi.org/10.5281/zenodo.7515107> and from E.F. upon reasonable request. Source data are provided with this paper.* »

With these improvements we trust that our manuscript now satisfies all points raised by the reviewers.

Yours sincerely,

Ellen Fogh, Niels B. Christensen and Rasmus Toft-Petersen

on behalf of the authors